# Influence of Brown or Germinated Brown Rice Supplementation on Fecal Short-Chain Fatty Acids and Microbiome in Diet-Induced Insulin-Resistant Mice

**DOI:** 10.3390/microorganisms11112629

**Published:** 2023-10-25

**Authors:** Ruozhi Zhao, Janice Fajardo, Garry X. Shen

**Affiliations:** Departments of Internal Medicine, Food and Human Nutritional Science, University of Manitoba, Winnipeg, MB R3E 3P4, Canada; zhaorz@yahoo.com (R.Z.); janice.fajardo@umanitoba.ca (J.F.)

**Keywords:** germinated brown rice, brown rice, high fat diet, mice, insulin resistance, gut microbiota, short chain fatty acids

## Abstract

Intake of whole grain foods is associated with improving metabolic profile compared to refined grain products, but the underlying mechanism remains unclear. The present study examined the effects of brown rice (BRR) or germinated brown rice (GBR) supplementation on fecal short-chain fatty acids (SCFAs), and relationship with gut microbiota, metabolism and inflammation in high fat (HF)-diet-fed mice. The results demonstrated that an HF diet supplemented with BRR or GBR comparably increased the abundance of fecal isobutyric acid compared to that in mice receiving HF+white rice (WHR) diet (*p* < 0.01). The abundance of valeric acid in HF+GBR-diet-fed mice was higher than those receiving HF+WHR diet (*p* < 0.05). The abundances of fecal isobutyric acid negatively correlated with fasting plasma glucose, insulin, cholesterol, triglycerides, tumor necrosis factor-α, plasminogen activator inhibit-1, monocyte chemotactic protein-1 and homeostatic model assessment of insulin resistance (*p* < 0.01). The abundance of valeric acids negatively correlated with insulin resistance (*p* < 0.05). The abundances of isobutyric acid positively correlated with *Lactobacillus*, but negatively correlated with *Dubosiella* genus bacteria (*p* < 0.05). The findings demonstrated that the increases in SCFAs in the feces of BRR and GBR-treated mice were associated with improvements in gut microbiome, metabolic and inflammatory profile, which may contribute to the antidiabetic and anti-inflammatory effects of the whole grains in HF-diet-fed mice.

## 1. Introduction

Type 2 diabetes (T2D) has become a worldwide epidemic chronic disease during the last 30 years. Diabetic complications lead to increased mortality, morbidity and socioeconomic burden in most developed or developing countries [1]. Recent studies indicated that unhealthy diet contributed to 70% of incident T2D in the world between 1990 and 2018. The top two attributors to inappropriate diet-induced T2D were the insufficient consumption of whole grain (26.1%) and the excessive intake of refined crop food products (24.6%) [2]. The underlying mechanism for the protective effects of whole grains on the development of T2D remains unclear.

Rice is the second most popular crop in the world, and it is the staple food for South and Southeast Asians, Middle Easterners and Latin Americans [3]. The most popularly consumed rice product is white rice (WHR), but the refining process removes the majority of nutrients from the whole grain product of rice, brown rice (BRR). Previous studies demonstrated that intake of BRR or germinated brown rice (GBR) reduced circulatory levels of glucose and lipids in human or experimental animal models [4,5,6]. Growing lines of evidence suggest that gut microbiota may play important roles in the development and treatment of chronic metabolic or inflammatory diseases, including diabetes [7]. A group of gut bacteria are capable of digesting insoluble fiber and producing short-chain fatty acids (SCFAs) in the intestinal tract. SCFAs are actively involved in the metabolism of glucose, lipids and the development of inflammation [8]. Limited information is available on the effects of whole grain products on the production of SCFAs. Previous studies found that BRR or GBR altered the generation of SCFAs in comparison to WHR or a regular diet in animals or humans, but the results were inconsistent [9,10,11]. Our recent studies demonstrated crosstalk between gut microbiota, metabolism and inflammation in mice receiving a high fat (HF) diet supplemented with WHR, BRR or GBR [12]. The roles of SCFAs in the beneficial effects of BRR and GBR in mice and their relationships with gut microbiota remain to be clarified.

The present study examined the influence of BRR and GBR versus WHR supplemented in an HF diet on the abundances of SCFAs in mice, and associations between fecal SCFAs and gut microbiota circulatory metabolic or inflammatory markers were further investigated. The outcome of the present study is expected to demonstrate the effects of BRR and GBR on the production of SCFAs in the intestine, and its relationship with gut microbiota, metabolism and chronic inflammation in HF-diet-fed mice.

## 2. Methods

### 2.1. Rice and Pre-Germination

BRR and refined WHR produced from the same batch of Qinpu Xingxiu short rice were obtained from Shanghai Chiming Scientific Technological Inc. (Shanghai, China). The pre-germination of BRR and chemical compositions of BRR, GBR and WHR were conducted as described previously [12]. Briefly, BRR was soaked in water for 8 h at 30 °C and then was submerged in a thin layer of water for 24 h at 30 °C. Sprouts of GBR were allowed to grow to 0.5–1 mm in length and then incubated in an oven for 10 min at 80 °C to terminate the pre-germination. GBR was dried at 50 °C for 8 h and then cooled down at room temperature before storage. WHR, BRR and GBR were mechanically milled, and the rice powders were passed through a 0.5 mm mesh.

### 2.2. Diet Composition

Carbohydrate-free HF diet powder (cat#: D12492px11) was received from Research Diets (New Brunswick, NJ, USA). Three types of diet were prepared by supplementation of 26 g% weight of different rice powders into carbohydrate-free HF diet powder then pelleted and dried. Each of the rice-supplemented HF diets contained 35 g% of fat (32 g% of lard, 3 g% of soybean oil), representing 60% of the total calories in the diet, 26 g% protein for 20% of calories and 26 g% carbohydrate (rice powder) for another 20% of calories in the diet.

### 2.3. Animals and Dietary Regimens

C57 BL/6J mice (male, 6 weeks of age) were obtained from Jackson Laboratory (Bar Harbor, ME, USA). Animals were held in stainless-steel cages in an air-conditioned room with routine maintenances and received regular rodent chow and tap water for 1 week. Following the stabilization, mice received an HF diet supplemented with WHR, BRR or GBR powder (n = 5/group) for 12 weeks.

### 2.4. Animal Monitoring and Sample Collection

Food intakes and body weights were measured at baseline and every two weeks across the dietary experiment. Feces were collected from mice individually caged overnight two days before the end of the regimen and stored at −70 °C before analysis. The mice fasted overnight the day before the end of the experiment. The mice were euthanized by inhaling 5% isoflurane followed by final blood withdrawal via cardiac puncture. The protocols of the animal experiments were approved by the Animal Management and Protocol Committee at the University of Manitoba.

### 2.5. Biochemical Measurements

The levels of fasting plasma glucose (FPG), cholesterol and triglycerides were analyzed using Sekisui Diagnostics SL reagents (Charlottetown, PE, Canada). Plasma insulin, plasminogen activator inhibitor-1 (PAI-1), monocyte chemotactic protein-1 (MCP-1) and tumor necrosis factor-α (TNFα) were assessed by enzyme-linked immunosorbent assays using reagent kits for mouse insulin (EMD, Millipore, Billerica, MA, USA), PAI-1 and TNFα (Thermo Fisher Scientific, Ottawa, ON, Canada) or MCP-1 (Abcam, Cambridge, MA, USA). Homeostatic model assessment of insulin resistance (HOMA-IR) was calculated from plasma glucose and insulin in simultaneously collected blood samples after fasting as described [13].

### 2.6. Fecal Bacterial DNA Extraction and Sequencing

Mouse fecal DNA was extracted using QIAGEN PowerFecal DNA Isolation Kit (Hilden, Germany) and quantified using a NanoDrop 2000 spectrophotometer (Wilmington, DE, USA). Bacterial DNA was amplified using polymerase chain reaction (PCR) with the following primers targeting the V4-V5 region of bacterial DNA: 515F (50-GTGYCAGCMGCCGCGGTAA) and 926R (50-CCGYCAATTYMTTTRAGTTT). The PCR amplicons were normalized by using Just-a-Plate 96-well normalization kit (Charm Biotech, Cape Girardeau, MO, USA), and then they were constructed into a library for 16S-rRNA gene sequencing on an Illumina MiSeq platform (San Diego, CA, USA) in the Integrated Microbiome Resource at Dalhousie University [14].

### 2.7. Measurement of Fecal SCFAs

Mouse fecal SCFAs were extracted using propyl chloroformate and derivatized with a reaction system containing water, propanol and pyridine, and the analyses were performed using an Agilent 7890A (Santa Clara, CA, USA) gas chromatography coupled with an Agilent 5975A inert XL EI/CI mass spectrometry in Microbiome Insights (Vancouver, BC, Canada) through customer service as previously described [15].

### 2.8. Bioinformatics Analysis and Statistics

Raw data of gut microbiota in the form of FASTQ files were demultiplexed according to barcode sequences, followed by trimming using Cutadapt (version 1.17), and analyzed on the Quantitative Insights Into Microbial Ecology 2 (QIIME2) platform. Diversities metrics (α- and β-diversity) of gut microbiota were assessed using QIIME2 as described [11]. Graphs for taxonomy and correlation were prepared using OriginLab Pro 2023b software. Liner discriminant analysis Effect Size (LEfSe) of gut microbiota was analyzed using Galaxy module [12]. Quantitative data were presented as means ± standard deviation (SD). Data from multiple groups were analyzed using analysis of variance (ANOVA) followed by Tukey’s test post hoc using OriginPro 2023 software. Collections between variables were assessed using linear regression analysis with SigmaPlot Window 10 software. Differences with probability < 0.05 were considered as significant.

## 3. Results

### 3.1. Influence of BRR and GBR on Fecal SCFAs

The abundance of fecal isobutyric acid in mice treated with an HF diet supplemented with 26 g% of BRR or GBR was significantly higher than that in mice treated with an HF + 26% of WHR (*p* < 0.01, Figure 1D). The levels of valeric acid in HF+GBR-, but not HF+BRR-, diet-fed mice were significantly higher than those in mice receiving HF+WHR diet (*p* < 0.05, Figure 1E). No significant difference in fecal acetic acid, propionic acid, butyric acid, isovaleric acid or hexanoic acid was detected among the three groups of mice (Figure 1).

### 3.2. Effects of BRR and GBR on Biochemical Variables, Body Weights and Food Intake

The levels of FPG, cholesterol, triglycerides, insulin and HOMA-IR in mice receiving an HF diet supplemented with BRR or GBR were significantly lower than those in mice fed with an HF diet supplemented with the equal dosage of WHR (*p* < 0.01). No significant difference in the metabolic variables was detected between mice receiving HF+BRR and HF+GBR diet (Figure 2A–D). Body weight and food intake were not significantly different among the three groups (Figure 2E,F).

### 3.3. Effects of BRR and GBR on Pro-Inflammatory Mediators

The levels of circulatory TNFα, PAI-1 and MCP-1 in mice treated with an HF diet supplemented with BRR or GBR were significantly lower than that in mice receiving HF+WHR diet (*p* < 0.01). No significant difference in the inflammatory markers was detected between mice treated with HF+BRR and HF+GBR diet (Figure 3A–C).

### 3.4. Correlations between Fecal SCFAs, Food Intake and Biochemical Variables

The abundance of acetic acid positively correlated with food intake, but negatively correlated with the levels of triglycerides, MCP-1 and TNFα in plasma (*p* < 0.05). The levels of isobutyric acid negatively correlated with plasma triglycerides, cholesterol, MCP-1, PAI-1, TNFα, glucose, insulin and HOMA-IR in the mice (*p* < 0.01). The levels of butyric acid negatively correlated with MCP-1 and PAI-1 (*p* < 0.01). The abundance of valeric acid negatively correlated with insulin and HOMA-IR in the mice (*p* < 0.05, Figure 4).

### 3.5. Influence of BRR and GBR on Gut Microbiota

BRR or GBR supplementation significantly increased the Shannon index α-diversity of gut microbiota (Figure 5A). The principal coordinate analysis plot demonstrated that the gut microbiota in mice fed with HF+BRR or HF+GBR diet was completely separated from that in the HF+WHR group. Partial overlap in the β-diversity of gut microbiota in mice receiving HF+BRR and HF+GBR diet was detected as expected (Figure 5B).

The abundances of fecal *Bacteroidates* phylum bacteria in mice fed with an HF diet supplemented with BRR or GBR were significantly higher than those in HF+WHR-diet-treated mice (*p* < 0.05). The levels of *Verrucomicrobia* phylum bacteria in mice receiving HF+BRR, but not HF+GBR, diet were significantly lower than those in HF+WHR-fed mice (*p* < 0.05, Figure 5C). The abundance of *Akkermansiaceae* family bacteria in mice fed with HF+BRR diet was significantly lower than that in mice receiving HF+WHR diet (*p* < 0.05). The abundance of *Lachnospiraceae* family bacteria in mice fed with an HF diet supplemented with BRR or GBR was significantly higher than that in mice fed with HF+WHR diet (*p* < 0.05, Figure 5D). The abundance of *Akkermansia* genus bacteria in mice receiving HF+BRR diet was significantly lower than that in mice fed with HF+WHR diet. The abundance of *Bacteroides* genus bacteria in HF+GBR-diet-treated mice was significantly greater than that in HF+WHR-diet-fed mice (*p* < 0.01). The abundances of *Dubosiella* genus bacteria in mice treated with diet containing BRR or GBR were significantly lower than those in mice receiving HF+WHR diet (*p* < 0.05 or 0.01). The abundances of *Lachnoclostridium* and *Lactococcus* genus bacteria were significantly higher in the feces of mice receiving HF+BRR diet (*p* < 0.05) compared to HF+WHR diet (*p* < 0.05, Figure 5E). The abundance of *Dubosiella newyorkensis* species bacteria in the feces of mice fed with HF+BRR or HF+GBR diet was significantly lower than that in mice receiving HF+WHR diet (*p* < 0.001). The abundance of *Lactococcus lactis* species bacteria in mice fed with HF+BRR diet was significantly higher than that in HF+WHR-diet-treated mice (*p* < 0.05, Figure 5F).

The results of LEfSe analysis displayed a distinguishable pattern of gut microbiota in mice who received an HF diet supplemented with different rice products. The gut microbiota of HF+WHR-diet-fed mice was characterized by the enrichment in *Erysipelotrichaceae*, *Dubosiella*, *Akkermansia* genus, *Akkermansiaceae* family and *Verrucomicrobia* phylum bacteria. Increased abundances of *Bacteroidates* phylum, *Bacteroidaceae* family and *Bacteroides* genus bacteria were characterized in mice receiving HF+GBR diet. The enrichment of *Lachnospiraceae* family bacteria and *Lactococcus* genus bacteria was characterized in mice fed with HF+BRR diet (Figure 5G,H).

### 3.6. Correlations between Fecal SCFAs and Microbiota

The following associations between fecal SCFAs and gut bacteria were detected. The abundances of fecal isobutyric acid negatively correlated with those of *Actinobacteria* and *Verrucomicrobia* phylum bacteria in the mice (*p* < 0.05 or 0.01). The abundance of butyric acid negatively corrected with fecal *Verrucomicrobia* (*p* < 0.05, Figure 6A). The levels of acetic acid positively correlated with *Lactobacillaceae* family bacteria (*p* < 0.05). The abundances of isoburyric acid positively correlated with *Lachnospirceae*, but negatively correlated with *Akkermansiaceae* and *Erysipelotrichaceae* family bacteria (*p* < 0.05). The levels of butyric acid negatively correlated with fecal *Akkermansiaceae* family bacteria. The abundance of valeric acid positively correlated with *Lachnospiraceae*, but negatively correlated with *Erysipelotrichaceae* family bacteria (*p* < 0.05). The abundance of hexanoic acid positively correlated with *Peptococcaeae* family bacteria (*p* < 0.01, Figure 6B). The levels of fecal isobutyric acid positively correlated with fecal *Lactobarcillus* and negatively correlated with *Dubosiella* genus bacteria (*p* < 0.05). The abundance of butyric acid negatively correlated with *Akkermansia* genus bacteria (*p* < 0.01). The abundances of valeric acid were negatively correlated with *Dubosiella* genus bacteria (*p* < 0.05). The abundance of hexanoic acid positively correlated with *Akkermansia* genus bacteria (*p* < 0.01, Figure 6C). The abundance of isobutyric acid negatively correlated with *Dubosiella newyorkensis* species bacteria in mouse feces (*p* < 0.01). The abundance of acetic acid positively correlated with *Lactobacillus gasseri*, a species of commonly used probiotic bacteria (*p* < 0.05, Figure 6D).

## 4. Discussion

The results of the present study demonstrated the following novel findings: (i) supplementation of BRR and GBR induced similar intensities of increase in isobutyric acid in the feces of HF-diet-fed mice compared to WHR; (ii) significantly increased fecal valeric acid was detected in mice receiving HF+GBR, but not HF+BRR, diet compared to WHR; iii) the abundances of fecal isobutyric, valeric, butyric, acetic acid or hexanoic acid negatively correlated with gut *Actinobacteria*, *Actinobactericeae*, *Verrucomicrobia*, *Erysipelotrichaceae*, *Akkermansia*, *Dubisiella* or *Dubosiella newyorkensis*, but positively correlated with *Lachnospiraceae*, *Lactobacillaceae*, *Peptococcaceae*, *Akkermansia*, *Lactobacillus* or *Lactobacillus gasseri*; iv) the abundances of fecal acetic acid negatively correlated with plasma triglycerides, MCP-1 and TNFα, but positively correlated with the food intake of mice. The levels of isobutyric acid in feces negatively correlated with all tested metabolic and inflammatory variables and positively correlated with food intake. The abundance of butyric acid negatively correlated with MCP-1 and PAI-1, and that of valeric acid negatively correlated with plasma insulin and HOMO-IR. The findings suggest that the levels of fecal SCFAs in mice were influenced by supplementation of BRR or GBR to an HF diet compared to WHR, and were correlated with changes in gut microbiota, metabolic or inflammatory markers in mice.

Our previous studies demonstrated that BRR and GBR supplementation comparably reduced FPG, lipids, insulin resistance and inflammatory markers in HF-diet-induced obese mice, and that was associated with changes in gut microbiota. Limited studies have examined the effects of whole grain rice products on both gut microbiota and SCFAs. An earlier study by Bird et al. reported that BRR increased fecal SCFAs in the large bowel of pigs [9]. A more recent study by Akamine et al. found that consumption of fermented brown rice beverage for 4 weeks significantly increased the abundance of *Sutterella wadsworthensis* in feces, which negatively correlated with fasting blood glucose, in patients with metabolic syndrome compared to that in patients receiving the WHR beverage, but no significant difference was detected in plasma SCFAs between patients who had the two different types of rice beverages [10]. Ding et al. (2022) demonstrated that an intake of GBR for 12 weeks decreased fast blood glucose in T2D patients compared to a regular diet, which was associated with increases in the abundances of fecal *Bacteroidetes* and multiple types of SCFAs [11]. The results from the present study support the findings by Bird et al. [9] and Ding et al. [11].

The results of the present study support the previous studies that whole grain rice products may improve metabolism and inflammatory status in HF-diet-fed mice, which was associated with increases in the abundances of beneficial gut bacteria and SCFAs in the gut. The results of the present study demonstrated that both BRR and GBR supplementation significantly increased the abundance of isobutyric acid in the feces of HF-diet-fed mice. Only GBR, but not BRR, supplementation augmented the abundance of fecal valeric acid in mice receiving an HF diet. The abundances of several fecal SCFAs correlated with the metabolic or inflammatory variables of the mice. For example, acetic acid positively correlated with probiotic *Lactobacillaceae* and fecal *Lactobacillus gasseri*, and negatively correlated with inflammatory markers in the peripheral circulation in the mice. Fecal butyric acid negatively correlated with pro-inflammatory MCP-1 and PAI-1 in the circulation of the mice. The abundance of fecal isobutyric acid negatively correlated with FPG, insulin resistance, lipids and inflammatory markers. The abundance of valeric acid negatively correlated with insulin and insulin resistance. The findings suggest that the changes in the composition of fecal SCFAs may be a part of the dietary responses of gut microbiota to the digestion of whole grain rice, which potentially contribute to the metabolic and anti-inflammatory benefits of BRR and GBR in mice on an HF diet.

Fecal isobutyric acid is primarily generated from gut microbiota through the degradation of branched amino acids (leucine, isoleucine, valine) or proline [16]. Isobutyric-acid-enhanced insulin stimulated glucose uptake and inhibited lipogenesis in adipocytes [17]. Fecal acetic acid may originate from foods or through the fermentation of undigested fiber by gut bacteria. Elevated production of acetic acids may increase insulin sensitivity and reduce lipolysis and inflammation in adipose tissue [18]. Valeric acid can promote glucose uptake in adipocytes and reduce insulin resistance [19]. The combination of the findings from the present study suggests that BRR and GBR may modulate the profile of bacteria in the intestine and further increase the production of SCFAs in the gut of HF-diet-fed mice. SCFAs may be involved in the regulation of glucose and lipid metabolism and attenuate inflammation, which contributes to the metabolic and anti-inflammatory benefits of whole grain food products.

Previous studies from our group and others demonstrated the prebiotic effect of GBR or BRR with increase in *Lactobacillus gasseri*, *Lactobacillus delbrueckii* subsp. *lactis* in in vivo and in vitro models [12,20]. The results of the present study further provide additional evidence for crosstalk between SCFAs and probiotic bacteria in the gut of mice treated with an HF diet supplemented with whole grain products.

BRR has noticeable health benefits, but its coarse texture, smell and difficult cooking process prevent it from becoming a popular rice product [21]. The production of GBR originally aimed to improve the taste and longer cooking time of BRR. The health benefits of GBR on glucose and lipid metabolism have been described in previous reports in animal models [12,22] and human subjects [6]. The results of the present study suggest that SCFAs generated from gut bacteria may contribute to the antidiabetic and anti-inflammatory effects of BRR and GBR. These findings may promote the consumption of GBR or BRR in one’s daily intake to replace or partially replace WHR for the prevention of diabetes, hyperlipidemia and chronic inflammation. The effective dosage of BRR or GBR consumption in humans still requires further investigation.

There are limitations to the present study: (i) only male mice were included in the present study due to the consideration that the global prevalence of T2D is higher in males in general [23]; subsequent studies in female animals may be required for the expanded application of the generated knowledge; (ii) the present study was only conducted in mice; further studies are needed in other species of animal models and clinical trials in human healthy subjects or patients with pre-diabetes or metabolic syndrome; (iii) BRR and GBR contain numerous compounds in addition to fiber. The identification of active compounds in BRR or GBR responsible for the changes in gut microbiota and SCFAs may help to further define the structure–function mechanism of benefits from BRR and GBR.

In conclusion, the findings of the present study demonstrated that the supplementation of BRR or GBR increased the production of selected types of SCFAs in distinguishable patterns in the gut of HF-diet-fed mice compared to WHR. The increases in SCFAs correlated with compositional changes in gut microbiota responded to BRR or GBR intake. That was associated with improved glucose and lipid profile, attenuated insulin resistance and inflammatory markers in mice. Promotion of the consumption of GBR or BRR and reduced intake of refined WHR may help to prevent diabetes-associated metabolic and inflammatory status. It may also help prevent further diabetic complications through the regulation of gut–organ communications and the upregulation of SCFAs.

## Figures and Tables

**Figure 1 microorganisms-11-02629-f001:**
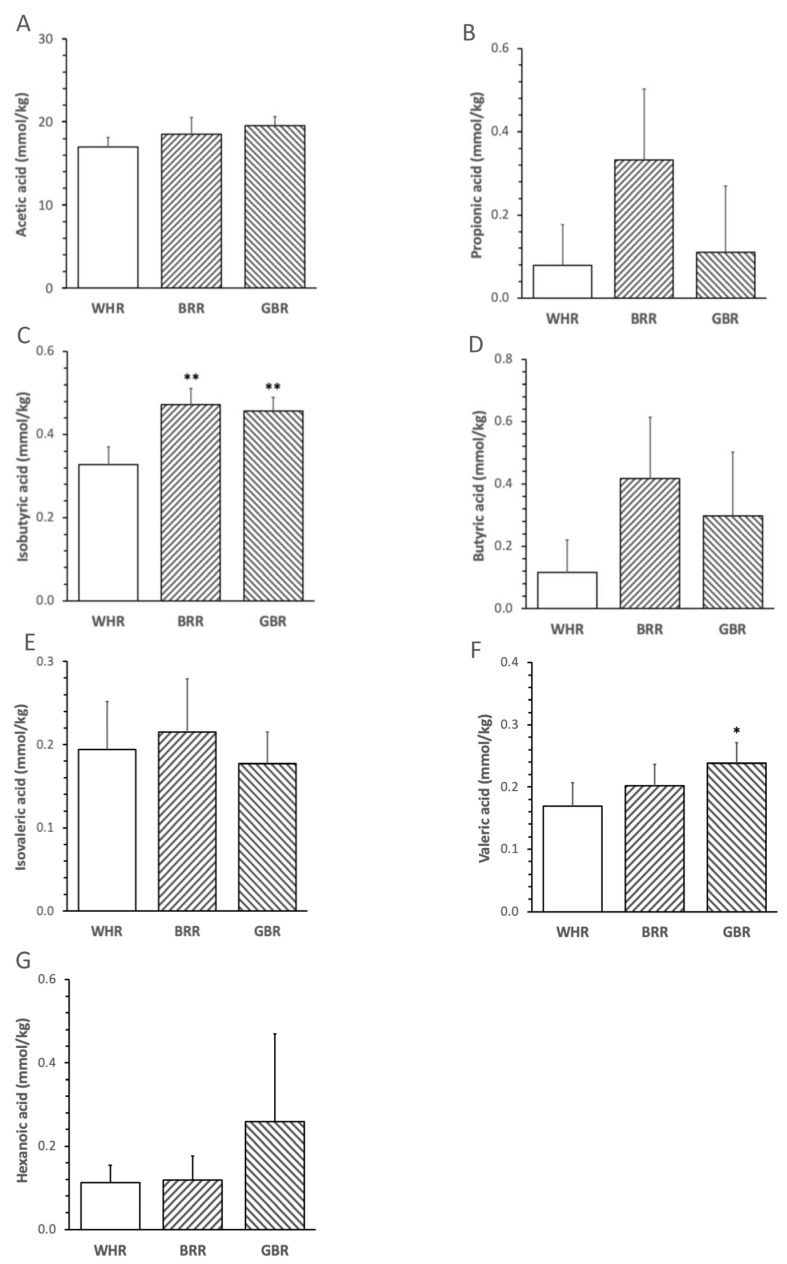
Effects of BRR and GBR on fecal short-chain fatty acids (SCFAs) in HF-diet-fed mice. Male mice (6 weeks of age,) were fed with HF diet supplemented with 26 g% of refined white rice (WHR), BRR or GBR for 12 weeks. SCFAs in feces were analyzed as described in the Methods section. (**A**): acetic acids; (**B**): propionic acids; (**C**): isobutyric acids; (**D**): butyric acids; (**E**): isovaleric acids; (**F**): valeric acids; (**G**): hexanoic acids. Values are expressed in means ± SD mmol/kg (n = 5/group). *, **: *p* < 0.05 or 0.01 versus WHR group.

**Figure 2 microorganisms-11-02629-f002:**
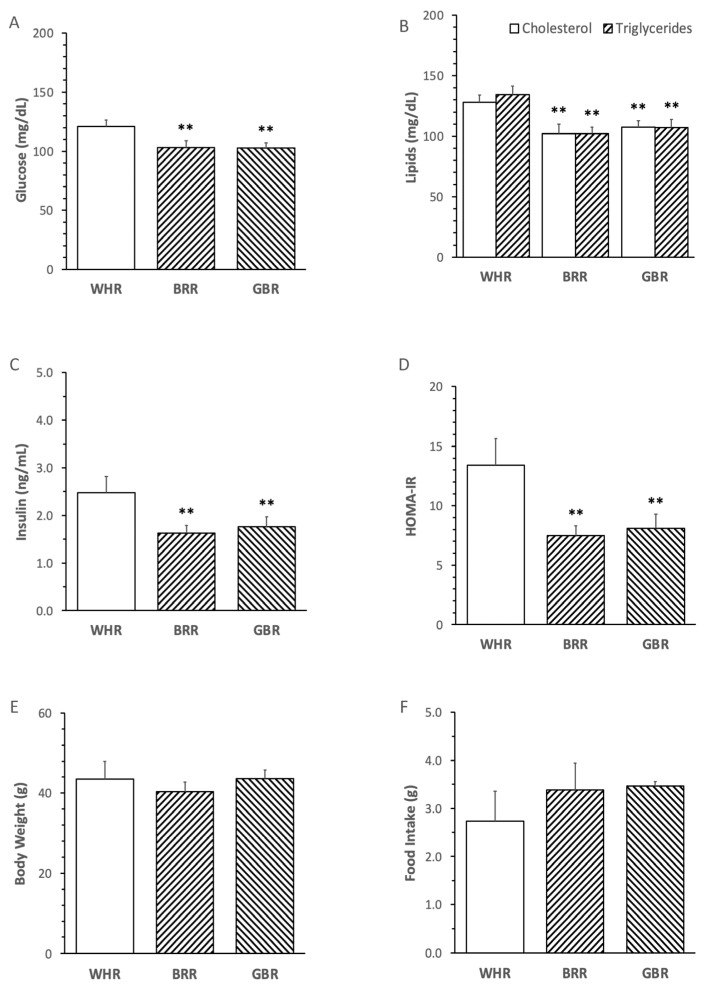
Effects of brown rice (BRR) and germinated brown rice (GBR) supplementation on metabolic variables in high fat (HF)-diet-fed mice. The dietary regimen of the animal experiment and sample collection procedure were identical to that described in the legend of Figure 1. Peripheral blood samples were collected at the end of dietary experiment after overnight fasting for measuring plasma glucose (**A**), cholesterol, triglycerides (**B**) and insulin (**C**). Homeostatic model assessment for insulin resistance (HOMA-IR) was calculated based on fasting glucose and insulin in simultaneously collected blood samples (**D**). Body weight (**E**) and daily food intake (**F**) were weighed at one day before the end of dietary experiment. Values were expressed in means ± SD (n = 5/group). **: *p* < 0.01 versus WHR group.

**Figure 3 microorganisms-11-02629-f003:**
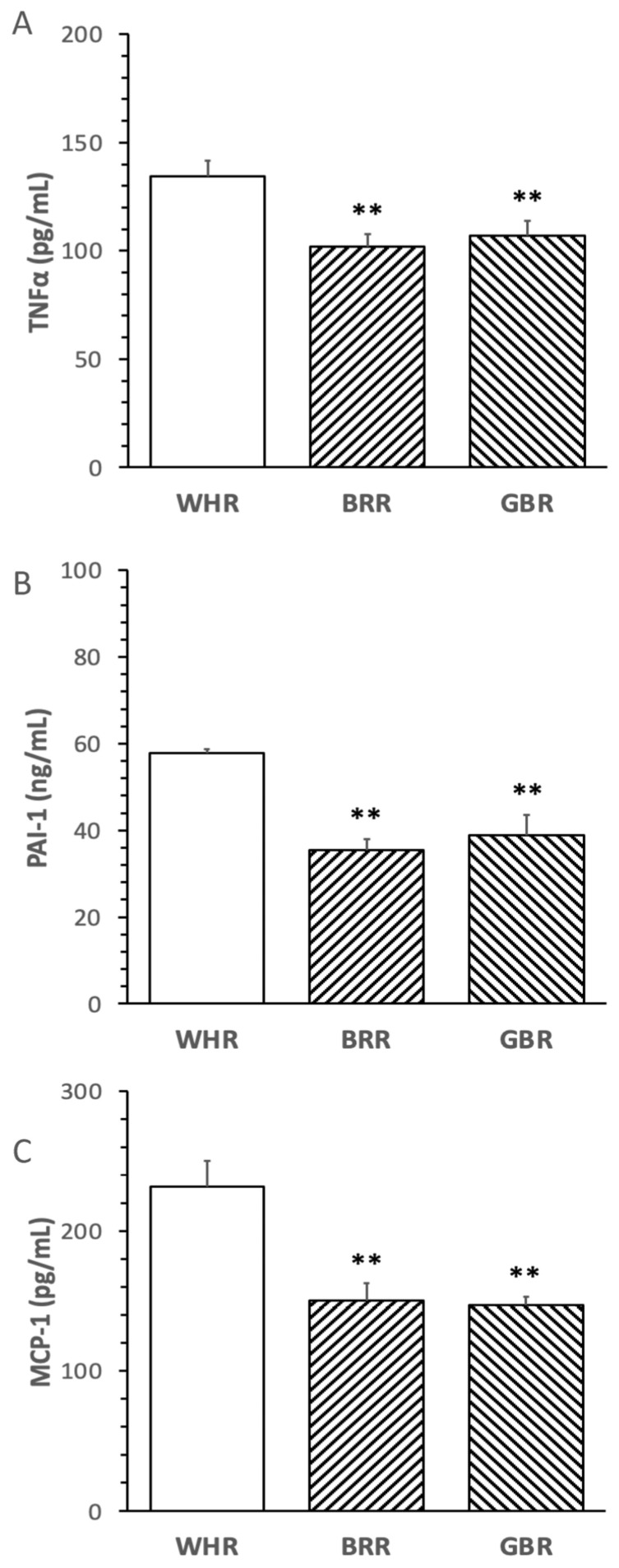
Influence of BRR and GBR on inflammatory markers in plasma of mice receiving HF diet. The dietary regimen of the animal experiment was identical to that described in the legend of Figure 1. The levels of inflammatory markers: (**A**) tumor necrosis factor-α (TNFα); (**B**) plasminogen activator inhibitor-1 (PAI-1); (**C**) monocyte chemotactic protein-1 (MCP-1) in plasma collected at the end of the dietary experiment were determined. Values were expressed in means ± SD (n = 5/group). **: *p* < 0.01 versus WHR group.

**Figure 4 microorganisms-11-02629-f004:**
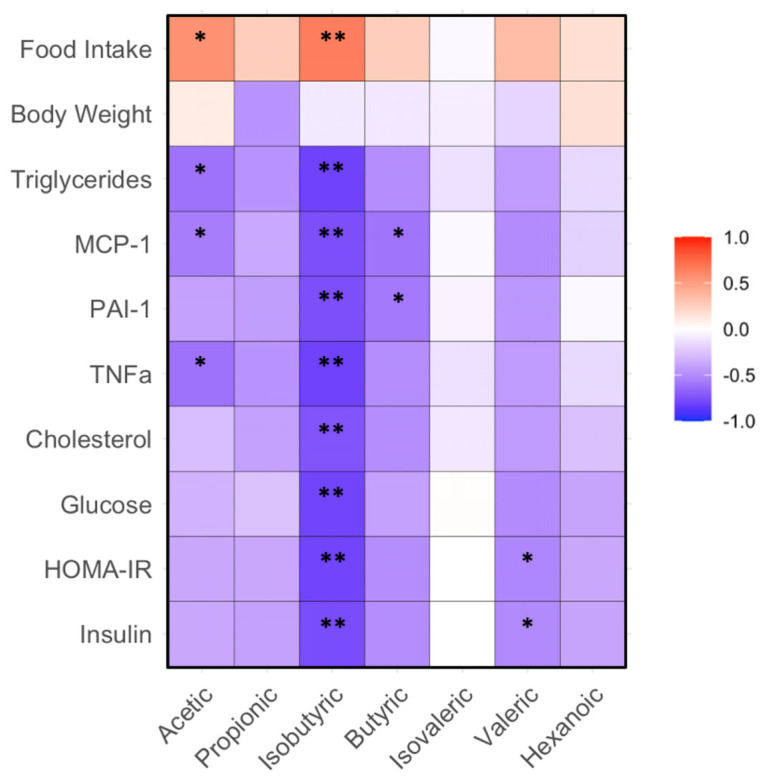
Correlation between gut bacteria and biochemical, body weight and food intake in mice fed with HF diet supplemented with BRR, GBR or WHR. The dietary regimen of the animal experiment and sample collection procedure were identical to that described in the legend of Figure 1. The sample analyses for gut microbiota and biochemical analysis were the same as described in the legends of Figure 1 and Figure 3. Red: positive correlation; blue: negative correlation. * or **, *p* < 0.05 or 0.01 for correlation efficient (r) between two variables (n = 15).

**Figure 5 microorganisms-11-02629-f005:**
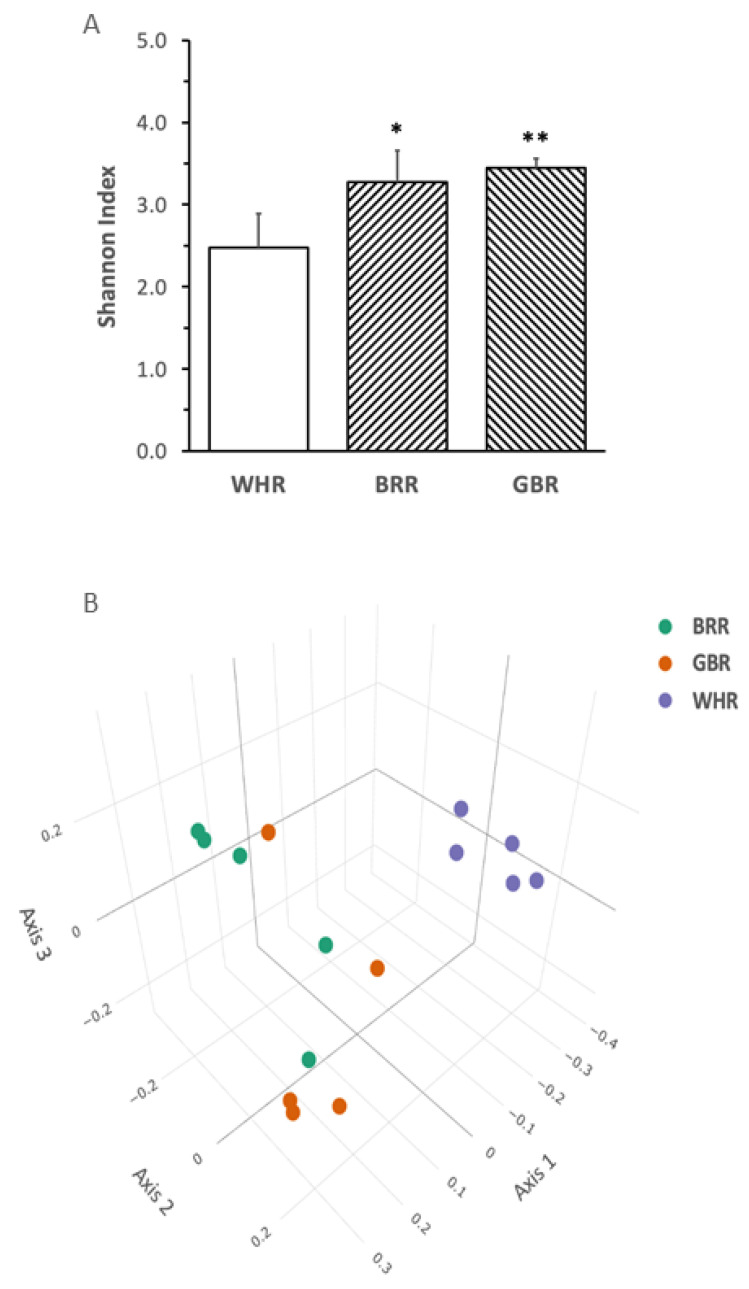
Effects of BRR and GBR on the composition of gut microbiota in HF-diet-fed mice. The dietary regimen of the animal experiment was identical to that described in the legend of Figure 1. Feces were collected from cages of individually hosted mice on last day of the dietary experiment. Fecal bacteria gene sequencing as described in the Methods section. (**A**): Shannon Index (α-diversity) values were expressed in means ± SD (n = 5/group), *, **: *p* < 0.05 or 0.01 versus WHR group; (**B**): principal component analysis (ß-diversity); (**C**): phylum bacteria, (**D**): family bacteria; (**E**): genus bacteria; (**F**): species bacteria; (**G**,**H**): liner discriminant analysis effect size of gut microbiota. Figure captions in Figure 5C–F: ANOVA: significance among the three groups; GBR-BRR, WHR-BRR, WHR-GBR: significance between the two groups using post-hoc test. *, ** or ***: *p* < 0.05, 0.01 or 0.001.

**Figure 6 microorganisms-11-02629-f006:**
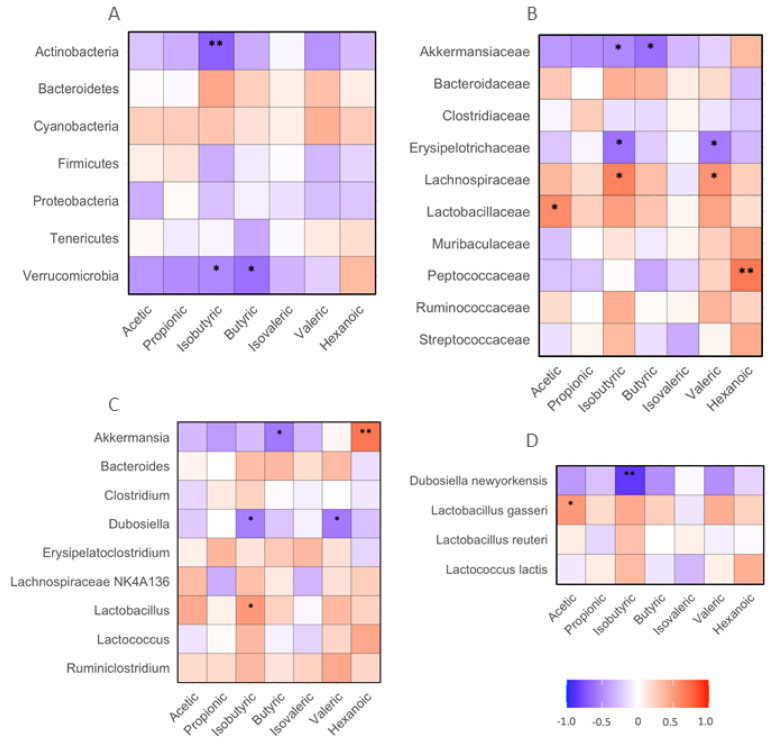
Correlation between SCFAs and biochemical, body weight and food intake in mice receiving HF diet supplemented with BRR, GBR or WHR. The dietary regimen of the animal experiment and sample collection procedure was identical to that described in the legend of Figure 1. The sample analyses for SCFAs and biochemical analysis were the same as described in the legends of Figure 1 and Figure 4. Red: positive correlation; blue: negative correlation. (**A**) phylum bacteria; (**B**) family bacteria; (**C**) genus bacteria; (**D**) species bacteria. *, **: *p* < 0.05 or 0.01 for correlation efficient (r) between two variables (n = 15).

## Data Availability

Not applicable.

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
