# Peer review of "Influence of Brown or Germinated Brown Rice Supplementation on Fecal Short-Chain Fatty Acids and Microbiome in Diet-Induced Insulin-Resistant Mice"

_microorganisms, 2023, doi:10.3390/microorganisms11112629_

Round 1

Reviewer 1 Report (New Reviewer)

In this manuscript of "Influence of Brown or Germinated Brown Rice Supplementation on Fecal Short Chain Fatty Acids and Microbiome in Diet-induced Insulin Resistant Mice", this research provides compelling evidence to support the beneficial effects of  brown rice (BRR) and germinated brown rice (GBR) in improving metabolic and inflammatory profiles in HF diet-fed mice. These improvements are linked to the increased levels of SCFAs, which are associated with changes in gut microbiome composition. A lot of work was done and it is contributing to the current scientific knowledge base in some way. But this MS needs revision and the author reply comments properly. The comments and questions are as follows:

1. The manuscript does not provide a detailed rationale or context for why this research is important or necessary. Including a brief statement about the significance of studying the effects of whole grains and SCFAs on metabolic health would strengthen the rationale.

2. Why did people not simply consume brown rice (BRR) or germinated brown rice (GBR) instead of going through the difficult process of obtaining refined white rice (WHR), which causes some of the benefits to be lost?

3. The figures in the manuscript are a bit blurry. Please consider replacing them with clearer ones.

4. Why are some of the data error bars in Figure 1 extremely large, while there are no significant differences?

Author Response

Responses to comments from Reviewer 1

Yes

Can be improved

Must be improved

Not applicable

Does the introduction provide sufficient background and include all relevant references?

( )

(x)

( )

( )

Are all the cited references relevant to the research?

( )

(x)

( )

( )

Is the research design appropriate?

( )

(x)

( )

( )

Are the methods adequately described?

(x)

( )

( )

( )

Are the results clearly presented?

( )

(x)

( )

( )

Are the conclusions supported by the results?

(x)

( )

( )

( )

Comments and Suggestions for Authors

In this manuscript of "Influence of Brown or Germinated Brown Rice Supplementation on Fecal Short Chain Fatty Acids and Microbiome in Diet-induced Insulin Resistant Mice", this research provides compelling evidence to support the beneficial effects of  brown rice (BRR) and germinated brown rice (GBR) in improving metabolic and inflammatory profiles in HF diet-fed mice. These improvements are linked to the increased levels of SCFAs, which are associated with changes in gut microbiome composition. A lot of work was done and it is contributing to the current scientific knowledge base in some way. But this MS needs revision and the author reply comments properly. The comments and questions are as follows:

Response: We appreciate the Reviewer for the comments on our manuscript and revision. WE will revise the manuscript again as suggested.

  1. The manuscript does not provide a detailed rationale or context for why this research is important or necessary. Including a brief statement about the significance of studying the effects of whole grains and SCFAs on metabolic health would strengthen the rationale.

Response: Thanks for the suggestion. We added a statement regarding the significance of the study in the introduction as suggested in the revised manuscript. 

  1. Why did people not simply consume brown rice (BRR) or germinated brown rice (GBR) instead of going through the difficult process of obtaining refined white rice (WHR), which causes some of the benefits to be lost?

Response: White rice is the most common product for rice and available in market worldwide Brown rice has a rough coat, which affects the taste and require longer time to cook. Germinate brown rice has relative soft coat, which improves the taste and shortens cooking time. Brown rice and germinated brown rice has limited demand and are not produced in large scale by industries, therefore their prices are more expensive than white rice so far. The cost and eating habit possibly are major reasons for white rice, instead of brown rice or germinated brown rice, is still the most popular rice product consumed in most regions worldwide. We added relevant information in the discussion in the revised version.

  1. The figures in the manuscript are a bit blurry. Please consider replacing them with clearer ones.

Response: Sorry for the quality of the figures. We will try our best to improve the quality of the figures.

  1. Why are some of the data error bars in Figure 1 extremely large, while there are no significant differences?

Response: The data of fecal short chain fatty acids (SCFA) were obtained through custom service. The large error bars possibly affect to distinguish their differences between groups. The size of animal groups probably contributes to the large error bars. However, we only considered SCFAs with significant difference between the groups as the effects of relevant rice products on the fecal production of SCFAs in high fat diet fed mice.

Reviewer 2 Report (Previous Reviewer 3)

The authors responded to the comments contained in my previous reviews and improved the manuscript. However, I believe that the authors should introduce a 'Conclusion' chapter into the manuscript, in which they should clearly formulate the most important conclusions from the conducted research.

Author Response

Response to the comments from Reviewer 2

Yes

Can be improved

Must be improved

Not applicable

Does the introduction provide sufficient background and include all relevant references?

(x)

( )

( )

( )

Are all the cited references relevant to the research?

(x)

( )

( )

( )

Is the research design appropriate?

(x)

( )

( )

( )

Are the methods adequately described?

(x)

( )

( )

( )

Are the results clearly presented?

(x)

( )

( )

( )

Are the conclusions supported by the results?

( )

(x)

( )

( )

Comments and Suggestions for Authors

The authors responded to the comments contained in my previous reviews and improved the manuscript. However, I believe that the authors should introduce a 'Conclusion' chapter into the manuscript, in which they should clearly formulate the most important conclusions from the conducted research.

Response: Thanks for the kind comments and suggestion. We added a conclusion paragraph at the end of discussion section as suggested.

This manuscript is a resubmission of an earlier submission. The following is a list of the peer review reports and author responses from that submission.

Round 1

Reviewer 1 Report

This manuscript is the research on " Influence of Brown or Germinated Brown Rice Supplementation on Fecal Short Chain Fatty Acids from High Fat Diet-Fed Mice: Relationship with Gut Microbiota and Circulatory Metabolic or Inflammatory Markers " submitted by Ruozhi Zhao, Janice Fajardo, Garry X. Shen *. The manuscript has some points should be improved.

1.      There is no figure or table information in the article

2.      The introduction of the article did not discuss in depth the relationship between different rice species and short-chain fatty acids

3.      Discussions in the article are not substantiated by citing appropriate research

Author Response

Responses to comments from Reviewer 1

  1. There is no figure or table information in the article

Response: All figures were submitted to Editorial Office. The authors are not sure the cause for the figure were not forwarded to the Reviewers. The resubmitted version including figures. 

  1. The introduction of the article did not discuss in depth the relationship between different rice species and short-chain fatty acids

Response: The authors appreciated the comments and added the discussion on the relationship between rice species and short chain fatty acids in the introduction of the revised manuscript.

  1. Discussions in the article are not substantiated by citing appropriate research

Response: The discussion of the revised manuscript has been substantially modified and relevant researches and references have been added.

Reviewer 2 Report

The manuscript investigated the influence of brown or germinated brown rice supplementation on fecal short chain fatty acids (SCFAs) of high fat diet-fed mice as well as the correlations between SCFAs, gut microbiota, circulatory metabolic or inflammatory markers, and food intake. It is an interesting study with some novelty and significance. However, I have some serious concerns about the research article and there are many points needed to be revised. The detailed comments are listed below.

1.      No figures were shown in the manuscript. Please put the figures in the manuscript where they were first mentioned. Moreover, please add line numbers in the manuscript.

2.       Please revise the title of this manuscript. Generally speaking, the words of the title are no more than 20 words. The prevent title is a little verbose.

3.      In the prevent abstract, there were too many descriptions about the results. Please focus on the most important results and the underlying mechanism. I suggest to rewrite the whole abstract.

4.      In the authors’ previous paper, the impact of germinated brown rice and brown rice on metabolism, inflammation, and gut microbiome in high fat diet-induced insulin resistant mice had already been reported (Zhao R, Huang F; Liu C, Asija V, Cao L, Zhou M, Gao H, Sun M, Weng XC, Huang J, Liao X, Liu Z, Sen L, Shen GX. Impact of germinated brown rice and brown rice on metabolism, inflammation and gut microbiome in high fat diet-induced insulin resistant mice. J Agri Food Chem 2022;70:14253-14246 ). Please assure that the data and figures shown in this manuscript have not been reported in the previous manuscript, especially the data of the effects of BRR and GBR on biochemical variables, body weights, food intake, pro-inflammatory mediators and gut microbiota.

5.      The underlying mechanism about the effect of brown or germinated brown rice on the high fat diet-fed mice and the relationship between gut microbiota and circulatory metabolic or inflammatory markers still needs to be further clearly amplified. It was a little too simple in the previous manuscript.

6.      The results and discussion were a little confusing and lack of logicality. I suggest to rewrite the two parts more logically.  

7.      References: the numbers of reference 3-23 are repeated. Please amend the number. Moreover, please unify the format of the journal name in the references, abbreviation or full writing.

8.      I suggest to add the part of “ABBREVIATIONS USED” in the end of this manuscript.

Minor editing of English language required.

Author Response

Reviewer 2

General comments: The manuscript investigated the influence of brown or germinated brown rice supplementation on fecal short chain fatty acids (SCFAs) of high fat diet-fed mice as well as the correlations between SCFAs, gut microbiota, circulatory metabolic or inflammatory markers, and food intake. It is an interesting study with some novelty and significance.

Response: The authors are grateful to the general comments to the manuscript.

The detailed comments are listed below.

  1. No figures were shown in the manuscript. Please put the figures in the manuscript where they were first mentioned. Moreover, please add line numbers in the manuscript.

Response: As indicated in the response to Reviewer 1, the authors surprised to hear that the manuscript sent to reviewers were not included. We submitted all figures with the manuscript in a signal file. It is true that figures were not inserted in the text as we selected not to submit in formatted file. This time, we submitted the text file inserted all figures as suggested.

Line numbers have been added in the revised manuscript.

  1. Please revise the title of this manuscript. Generally speaking, the words of the title are no more than 20 words. The prevent title is a little verbose.

Response: The title of the manuscript has been revised with acceptable length. As suggested.

  1. In the prevent abstract, there were too many descriptions about the results. Please focus on the most important results and the underlying mechanism. I suggest to rewrite the whole abstract.

Response: The abstract has been rewritten to focus of most important results and significance as suggested.

  1. In the authors’ previous paper, the impact of germinated brown rice and brown rice on metabolism, inflammation, and gut microbiome in high fat diet-induced insulin resistant mice had already been reported (Zhao R, Huang F; Liu C, Asija V, Cao L, Zhou M, Gao H, Sun M, Weng XC, Huang J, Liao X, Liu Z, Sen L, Shen GX. Impact of germinated brown rice and brown rice on metabolism, inflammation and gut microbiome in high fat diet-induced insulin resistant mice. J Agri Food Chem 2022;70:14253-14246 ). Please assure that the data and figures shown in this manuscript have not been reported in the previous manuscript, especially the data of the effects of BRR and GBR on biochemical variables, body weights, food intake, pro-inflammatory mediators and gut microbiota.

Response: Although the part of the design of the present study was similar to our previous report, but the data and figures in the present study have not been reported anywhere and the aim of short chain fatty acids (SCFAs) and the relationship between SCFAs and gut microbiome are novel in this line of study, We rearrange the results and figures to and remove the part of the correlations between gut microbiota and metabolic/inflammation data (Figure 7 in previous version) to emphasize the new findings from the revised manuscript.

  1. The underlying mechanism about the effect of brown or germinated brown rice on the high fat diet-fed mice and the relationship between gut microbiota and circulatory metabolic or inflammatory markers still needs to be further clearly amplified. It was a little too simple in the previous manuscript.

Response: The discussion on underlying mechanism for the effects of germinated brown rice on gut microbiome, SCFAs, metabolism and inflammation in high fat diet fed mice has been expanded as suggested in the revised manuscript.

  1. The results and discussion were a little confusing and lack of logicality. I suggest to rewrite the two parts more logically.

Response: The results and discussion parts have been rewritten as suggested.

  1. References: the numbers of reference 3-23 are repeated. Please amend the number. Moreover, please unify the format of the journal name in the references, abbreviation or full writing.

Response: The references have been reorganized and format of references were modified.

  1. I suggest to add the part of “ABBREVIATION USED” in the end of manuscript.

Response: A list of abbreviations used in the manuscript has been added and we would like to thank the Reviewer for the suggestion.

Reviewer 3 Report

The subject of the research is interesting and important from a medical point of view. However, the manuscript cannot be fully evaluated as the authors in the manuscript did not provide Figures from their research. The authors describe Figures, but they are not in the manuscript. In addition, authors should accurately describe the GC-MS method used. Authors must describe the validation of the GC-MS method used. The authors must present the conclusions of the conducted research.

Author Response

Comments from Reviewer 3 is not available.

Round 2

Reviewer 1 Report

I can't see where the corrections of article has been marked

The resolution of the added chart is seriously insufficient

Reviewer 3 Report

The presented second version of the manuscript is now available for review. The authors substantially corrected the manuscript, i.e. mainly supplemented it with missing figures. However, the quality of most drawings is poor. The authors did not respond to my first review. They did not include all my previous comments in the revised manuscript. Namely, authors should:
1) provide more details on the determination of short-chain fatty acids by GC-MS method;
2) present the validation of this GC-MS method for the determination of short-chain fatty acids;
3) present sample GC-MS chromatograms in the manuscript or supplementary materials.